# A Neurobiological Framework for the Therapeutic Potential of Music and Sound Interventions for Post-Traumatic Stress Symptoms in Critical Illness Survivors

**DOI:** 10.3390/ijerph19053113

**Published:** 2022-03-06

**Authors:** Usha Pant, Michael Frishkopf, Tanya Park, Colleen M. Norris, Elizabeth Papathanassoglou

**Affiliations:** 1Faculty of Nursing, Edmonton Clinic Health Academy (ECHA), University of Alberta, 11405-87th Ave, Edmonton, AB T6G 1C9, Canada; upant@ualberta.ca (U.P.); tmpark@ualberta.ca (T.P.); cnorris@ualberta.ca (C.M.N.); 2Department of Music, Faculty of Arts, University of Alberta, 3-98 Fine Arts Building, Edmonton, AB T6G 2C9, Canada; michaelf@ualberta.ca; 3Faculty of Medicine and Dentistry, University of Alberta, Walter C. MacKenzie Health Sciences Centre, Edmonton, AB T6G 2R7, Canada; 4Canadian Centre for Ethnomusicology (CCE), University of Alberta, 11204-89 Ave NW, Edmonton, AB T6G 2J4, Canada; 5School of Public Health, University of Alberta, ECHA 4-081, 11405-87 Ave NW, Edmonton, AB T6G 1C9, Canada; 6Cardiovascular Health and Stroke Strategic Clinical Network, Alberta Health Services Corporate Office Seventh Street Plaza 14th Floor, North Tower 10030-107 Street NW, Edmonton, AB T5J 3E4, Canada; 7Neurosciences Rehabilitation & Vision Strategic Clinical Network^TM^, Alberta Health Services Corporate Office Seventh Street Plaza 14th Floor, North Tower 10030-107 Street NW, Edmonton, AB T5J 3E4, Canada

**Keywords:** music, post traumatic stress disorder, critical illness, neurobiology, autonomic nervous system, limbic system

## Abstract

Overview: Post traumatic stress disorder (PTSD) has emerged as a severely debilitating psychiatric disorder associated with critical illness. Little progress has been made in the treatment of post-intensive care unit (ICU) PTSD. Aim: To synthesize neurobiological evidence on the pathophysiology of PTSD and the brain areas involved, and to highlight the potential of music to treat post-ICU PTSD. Methods: Critical narrative review to elucidate an evidence-based neurobiological framework to inform the study of music interventions for PTSD post-ICU. Literature searches were performed in PubMed and CINAHL. The Scale for the Assessment of Narrative Review Articles (SANRA) guided reporting. Results: A dysfunctional HPA axis feedback loop, an increased amygdalic response, hippocampal atrophy, and a hypoactive prefrontal cortex contribute to PTSD symptoms. Playing or listening to music can stimulate neurogenesis and neuroplasticity, enhance brain recovery, and normalize stress response. Additionally, evidence supports effectiveness of music to improve coping and emotional regulation, decrease dissociation symptoms, reduce depression and anxiety levels, and overall reduce severity of PTSD symptoms. Conclusions: Despite the lack of music interventions for ICU survivors, music has the potential to help people suffering from PTSD by decreasing amygdala activity, improving hippocampal and prefrontal brain function, and balancing the HPA-axis.

## 1. Introduction

Ongoing advancements in Intensive Care Unit (ICU) technology and evidence-based practice have significantly reduced ICU mortality. However, the intense stress and adverse emotions experienced during hospitalization in an ICU have long term effects on survivors’ physiological and psychological well-being [1,2,3]. Post Traumatic Stress Disorder (PTSD) has emerged as a major long-term complication of critical illness, along with depression and anxiety disorders [2,3,4,5]. Conservative estimates predict that one in every five survivors of critical care exhibit clinically significant symptoms of PTSD in the first year following ICU discharge [6]. Some variance in prevalence estimates exists, from 5 to 63% depending upon the time of assessment, screening tool and population [4,7,8,9,10,11]. The psychological impairments following critical illness constitute a significant public health issue [12]. Although integrative interventions and music therapy during the acute ICU stay have received some attention [13,14], there is a paucity of studies exploring the effects of music post-ICU discharge. This review seeks to raise awareness regarding this overlooked area, gather and report all the current literature on this topic, and to propose an evidence-based theoretical framework to guide music and sound interventions for ICU survivors.

PTSD is characterized by symptoms including: intrusive memories, hyper-arousal, avoidance of trauma-related stimuli, and negative alterations in mood or cognition that develop following exposure to traumatic life events. According to the Diagnostic and Statistical Manual of Mental Disorders, 5th Edition (2013) criteria for diagnosis, PTSD can be diagnosed if an individual is exposed to actual or threatened serious injury or death. Events such as critical illness, ICU admission and related ICU treatments, such as awareness during intubation, can meet the first criteria for PTSD [15]. The additional criteria include intrusive symptoms such as; (a) persistently re-experiencing the event through nightmares or flashbacks, (b) avoidance of trauma-related stimuli (they avoid remembering or thinking about the event and resist talking about how they feel about it), (c) new negative alterations in mood or cognition (depressed mood, trouble concentrating and irritability) and (d) hyperarousal (increase in arousal or anxiety that was not present prior to the trauma) that Toplasts more than 1 month and causes significant distress or changes in functionality [15]. PTSD affects cognitive abilities such as memory and learning, and often leads to social withdrawal. PTSD influences quality of life in ICU survivors, affecting their lives in intensely negative ways, including their ability to work, study and even carry out daily activities of living [2,16].

The combination of life-threatening illness or injury, along with the impact of potentially traumatic experiences in the ICU can contribute to the development of PTSD [7,17,18]. Potential traumatic experiences during critical illness are associated with awareness during painful procedures, such as intubation, sense of helplessness, hallucinations, loss of control and fear of death [19,20,21,22]. The specific ICU-related risk factors include in-hospital stress, ICU length of stay and mechanical ventilation, delirium, use of sedatives and analgesics, high disease severity and negative ICU experience [4,23,24]. Pre-existing psychopathology is the only pre-ICU factor associated with PTSD in the ICU survivors.

## 2. Aim

This review aimed to summarize neurobiological evidence on the pathophysiology of PTSD and the areas of the brain involved, as well as some of the effects of music on PTSD and ICU survivors to highlight potential mechanisms and effects of music on individuals suffering from post-ICU PTSD. Delineating an evidence-based theoretical framework of the effects of music for rehabilitation after critical illness can inform interventions that improve survivors’ psychological well-being and health outcomes.

## 3. Methods

We used a critical narrative review methodology as this was deemed most appropriate for the delineation of a theoretical framework on music effects on PTSD post-ICU. As there is a lack of guidelines for narrative reviews, we followed methodological recommendations by Ferrari [25]. We searched current research evidence on the pathophysiology of PTSD, especially pertaining to alterations in specific brain regions, and juxtaposed these findings with the neurobiological effects of music therapy in the brain, and its effects on neuropsychiatric conditions. We used a combination of search terms for music and PTSD to conduct keyword searches in PubMed and CINAHL databases spanning the time period from January 2000 to October 2021. The databases were searched individually for the following keyword combinations: (music or sound or audio or melody or playing or improvisation or song writing or singing or song or GIM or Guided Imagery and Music) AND (PTSD or post traumatic stress disorder or stress disorder).

Inclusion criteria included: systematic syntheses of evidence, primary research studies in humans regardless of methodology, studies that addressed the pathophysiology and neurobiology of PTSD, and studies that addressed the physiological effects of music on brain function. Studies that were not based on research evidence, that included non-human participants, and were not reported in English were excluded. Due to the scarcity of literature on the topic, we did not include any chronological criteria. We did not employ a formal approach to quality appraisal. We extracted and synthesized results of PTSD-related alterations on specific brain structures, and the impacts and mechanisms of music effects on these brain structures, as well as evidence on the effects of music on related neuropsychiatric entities. The Scale for the Assessment of Narrative Review Articles (SANRA) guided reporting [26].

## 4. Results

Our search strategy yielded 363 articles addressing the neurobiology of PTSD, 841 articles addressing the effects of music on PTSD and 61 articles on the neurobiological effects of music therapy in the brain.

### 4.1. Pathophysiology of PTSD

To summarize evidence on how the brain is affected by PTSD, it is essential to examine the key structures of the brain that are associated with how memories are stored, and how stimuli are associated with emotion. These key structures include: the prefrontal cortex (PFC), the amygdalae, and the hippocampus. Alterations in these areas of the brain are observed across various studies assessing PTSD [27,28]. Since the Hypothalamus-pituitary-adrenal axis (HPA axis) is the primary endocrine mediator of stress responses, studies have also illustrated the role of the HPA in the onset and perpetuation of PTSD [29,30,31]. A summary of studies that used imaging techniques to assess functional or structural brain alterations is presented in Table 1. 

#### 4.1.1. The Hypothalamus-Pituitary-Adrenal Axis

The HPA axis involves a complex set of interactions among the hypothalamus, the pituitary gland, and the adrenal gland that determines the level of circulating cortisol. Stress triggers an HPA and sympathetic nervous system (SNS) response. Upon perception of a stressful stimulus, norepinephrine and indirect limbic inputs from the hippocampus, prefrontal cortex, and amygdala stimulate neurons in the paraventricular nucleus (PVN) of the hypothalamus that contain a corticotropin releasing factor (CRF), leading to the activation of the HPA axis and the release of adrenocorticotropic hormone (ACTH) into the systemic circulation. ACTH then binds to melanocortin 2 receptors in the zona fasciculata of the adrenal cortex and stimulates the release of glucocorticoids (specifically cortisol). Activation of the HPA axis is modulated by pituitary adenylate cyclase-activating polypeptide (PACAP), which appears to mediate the production of CRH. PACAP is also involved in the modulation of the sympathetic nervous system (SNS) response [32].

The SNS contributes to the flight or fight response by signaling the adrenal medulla to release catecholamines (epinephrine and norepinephrine) and enkephalins. Cortisol, catecholamines and enkephalins together stimulate a series of effects such as enhancing glucose availability, regulating the immune system and brain function, and impacting electrolyte balance to manage stressors [31]. Simultaneously, several brain structures control the HPA axis activity. Specifically, both the hippocampus and PFC impede the CRF neurons in the PVN of the hypothalamus. In contrast, the amygdala triggers CRF neurons in the PVN. Cortisol regulates HPA-axis activity by generating negative feedback to both the hypothalamus and the anterior pituitary.

Cortisol acts as the primary molecule to enable the stress response, as well as prevent ongoing HPA axis activity. The function of the HPA axis is controlled by two factors: (a) the effectiveness and (b) number of glucocorticoid receptors in the pituitary and hypothalamus [29,33], which regulate both CRF and ACTH release. However, if the negative feedback cycle of the HPA axis is disrupted, either due to the overactivity of CRF or due to hypersensitivity to glucocorticoids, the production of cortisol continues. This negative feedback system appears to be compromised in patients with PTSD. A metanalysis of 24 studies examining six HPA-axis genes in PTSD patients demonstrated involvement of two genes: (a) NR3C1 associated with the encoding of the glucocorticoid receptor, and (b) FKBP5 linked with regulating the affinity of the glucocorticoid receptor [34]. Moreover, persistent exposure to stressful events leads to multiple such cycles in a single day, preventing the HPA axis from returning to the baseline [35]. The aberrant stress response resulting from an overactive and prolonged HPA axis response increases stress-like symptoms in people with PTSD [36]. Moreover, the over-production of cortisol generates a state of toxic stress that changes the physical structure and function of the amygdala, hippocampus, and PFC (Figure 1).

#### 4.1.2. The Amygdalae

The amygdalae are a subcortical collection of nuclei situated in the anterior temporal lobe of each hemisphere, projecting to the brainstem and hypothalamic regions. The amygdalae play a critical role in emotional processing and generation of fear responses. In particular, amygdalae are associated with the execution of the physical, autonomic, and musculoskeletal components of the emotional response. Moreover, they have connections with other emotional centers in the brain. The amygdalae process the stressful events resulting in the release of cortisol through HPA-axis activation. In the case of an aberrant HPA axis feedback cycle, the continuous cortisol release enhances the amygdala’s ability to communicate within and with other brain structures [36,38]. This makes the amygdalae more receptive to perceived threat. A metanalysis of fifteen functional imaging studies investigating PTSD patients demonstrated significant hyperactivation of the amygdalae [39].

The hyperactive amygdalae shift the brain’s processing mode from the thoughtful prefrontal cortex pathway to its own rapid, emotional, and reactive pathway [40,41]. Thus, the hyperactive amygdala promotes hypervigilance and impairs discrimination of threatening stimuli [36]. This is a mechanism involved in the increased hyperawareness to stimuli that are not even related to the trauma experience in individuals with PTSD.

#### 4.1.3. The Hippocampus

The hippocampus is an extension of the cerebral cortex situated deep within the temporal lobe. The hippocampus plays a crucial role in the consolidation of information from short-term memory to long-term memory. It is also involved in the neuroendocrine regulation of stress hormones. Alterations in the structure and function of the hippocampus are observed in several neurological and psychiatric disorders [36,42]. Boccia et al., (2016) through functional magnetic resonance imaging, observed a hypoactive hippocampus among participants with PTSD [27]. The hippocampus has projections to the hypothalamus and is involved in the regulation of adrenocorticotropic hormones. Therefore, hypo-activity of the hippocampus may result in increased HPA axis activity [43].

Traumatic stressors have also been shown to alter hippocampal dendritic morphology and inhibit neurogenesis in the hippocampus [42,44,45,46]. Chronic stress rapidly reduces the number of dendritic spines and branches of pyramidal neurons in the Cornu Ammonis subfield 3 (CA3) and compromises the integrity of CA1, which is involved in the persistence and re-experiencing of traumatic memories [47,48]. For example, early magnetic resonance imaging studies demonstrated smaller hippocampal volumes in Vietnam Veterans with PTSD compared with controls [49,50,51,52,53]. Small hippocampal volumes were associated with the severity of trauma and memory impairments in these studies. Stress also suppresses the production of new granule neurons in the dentate gyrus regions of the hippocampus [54]. Evidence also shows that small hippocampal volumes are involved in increased susceptibility to stress and trauma [55]. Moreover, prolonged HPA axis activation generates various neurobiological changes in the hippocampus that influence the hippocampal functions, such as learning and memory functioning [44].

Altogether, these changes can result in generalizing the fear from the traumatic event to learned-fear in situations that are not related to the traumatic event, with hypervigilance and avoidance behaviors. Thus, the person with PTSD has a difficult time distinguishing between safe and unsafe stimuli and re-experiences a physiological and emotional toll similar to the traumatic event. This results in hypervigilance [38], exaggerated stress response and inability to prevent other fear associations [41,56]. Moreover, the effects on the hippocampus also inhibit learning and recall of previously stored memories [57], contributing to repressed memories in PTSD patients.

#### 4.1.4. The Pre-Frontal Cortex

PFC is vitally involved in executive functions, such as concentration, organization, judgement, reasoning, problem solving, decision making, creativity, emotional regulation, and abstract thinking. Chronic exposure to stress impairs prefrontal cortex functioning, which leads to aberrant stress responses and maladaptive coping. Decreased volumes of the frontal cortex are associated with significant hypoactivation of the PFC in individuals with PTSD [39,58]. There are dense white matter connections between the ventral region of PFC and the amygdalae that facilitate bi-directional communication between these two areas. Amygdalic activity is inhibited through the PFC [59,60,61,62,63]. Therefore, a hypoactive PFC in individuals with PTSD may impair regulation of emotional processing in the amygdalae [64]. Besides this, prefrontal dysfunction also results in a reduced ability to concentrate and regulate executive functions [36,65]. Thus, the hypoactivity of the prefrontal cortex can explain some of the symptoms of PTSD such as the inability to focus, solve problems and guide thoughts or emotions using working memory [40,66]. A summary of neurobiological changes in hippocampus, amygdala and PFC and related effects on behavior is presented in Table 2.

### 4.2. Music Therapy

Music therapy is a systematic process of intervention wherein the therapist helps the client to promote health, using musical experiences, and the relationships that develop through them as dynamic forces of change [67]. Music not only can evoke feelings, but also engages and motivates people to connect to others and offers a medium of nonverbal communication [68]. Music therapy is a type of expressive arts therapy that uses music to improve and maintain the physical, psychological, and social well-being of individuals and involves a broad range of activities, such as listening to music, singing, and playing a musical instrument [68].

Active music therapy engages patients in some form of music-making, such as vocalizing, singing, playing instruments, song writing or composing, and conducting [68]. Receptive music therapy guides patients or clients in listening or responding to live or recorded music through dancing or other movement to music, or lyric discussion [68,69].

#### 4.2.1. Effect of Music in Neuropsychiatric Conditions and PTSD

Various music and sound interventions have been used to improve health outcomes in a broad spectrum of psychological and neurological disorders [70,71,72,73]. Music therapy has demonstrated significant effectiveness for the reduction of depression in people with dementia [71,74] and improvement of mobility in people with stroke [75]. A systematic review of music therapy delivered by a professional music therapist revealed the efficiency of music to improve social interaction and communication skills in children with autism spectrum disorder [76]. Moreover, there is evidence that music therapy in addition to standard care improved mental state and social functioning in schizophrenia patients [71,77].

A meta-ethnography of 46 qualitative studies found that participatory music engagement, music actively made by the participant, including singing, and not limited by musical genre such as classical or jazz, improved well-being by facilitating self-development, providing respite from problems, and fostering social connections [78]. There is growing evidence that music therapy can abate the stress response, decrease anxiety, and induce an overall relaxation response by reducing stress-inducing stimuli. A recent meta-analytic study by Witte et al. (2020) revealed the effectiveness of music interventions to relieve stress in a variety of settings, including mental health, polyclinic medical settings, medical surgery, and everyday life. The study’s findings demonstrated that pre-recorded relaxation music without lyrics could reduce physiological stress symptoms such as heart rate, blood pressure, and stress-related hormones, as well as psychological stress symptoms such as anxiety, nervousness, restlessness, and feelings of worry [79]. McKinney & Honig (2017), in a systematic review across populations, examined effects from randomized and non-randomized controlled trials and found a medium to large effect of guided imagery and western classical music on various psychological measures including anxiety, and mood disturbance [80]. A Cochrane review by Bradt et al. (2013) demonstrated the beneficial effect of patient-selected music from different styles of music such as jazz, easy listening, country and western, or classical music on preoperative anxiety and recommended its use as an alternative to sedative drugs [81]. Another review of the literature by Hole et al. (2015) confirmed that Chinese classical music reduced postoperative pain, anxiety, and analgesia use and increased patient satisfaction [82].

The effect of music and sound interventions has also been explored in ICU patients [13]. Various forms of music interventions in ICU populations are found beneficial in reducing ICU-related anxiety and in-hospital stress [13,14]. In mechanically ventilated patients, patient-directed music therapy is associated with lower anxiety scores, sedation frequency, and sedation intensity when compared to usual care [83,84,85,86]. A majority of included studies used music that contains simple repetitive rhythms, low pitch, slow tempos, harmony and lack percussive instruments and vocals. A literature review by Hetland et al. (2015) indicated that relaxing music, such as nature-based sounds, classical, and easy listening, can help manage pain, agitation, delirium, post-traumatic stress disorder (PTSD), anxiety, and depression in ICU patients by reducing the need for sedatives during mechanical ventilation, length of stay, and physiologic signs of anxiety and biomarkers of the stress response [87]. Moreover, implementing music interventions in usual care is free of adverse side effects and can also reduce ICU costs. A recent study by Chlan et al. (2018) demonstrated that patient-directed music interventions can save about $2000/patient and concurrently better manage anxiety with less sedative medication than usual care [88]. However, to date, the effect of music and sound interventions have not been explored in relation to the psychiatric disorders and PTSD after discharge from the ICU.

Despite this gap in evidence, music appears to be a promising adjunct in the treatment of PTSD [70]. Evidence from various studies shows a significant effect of individual and group music therapy in the reduction of core PTSD symptoms and the increase in social function among PTSD patients (Table 3). A systematic review on creative art therapy also pointed out the potential of relaxation music therapy to creatively process, cope, and recover from PTSD [89]. A mixed method study by Story and Beck (2017) reported experiencing classical music as a tool for coping with PTSD symptoms, particularly to regulate emotions, decrease arousal, express repressed feelings, and connect with others [90]. Moreover, group music therapy in adult psychiatric patients with persistent PTSD, who had been unable to benefit from cognitive behavioral therapy CBT, showed a significant decrease of all dimensions of PTSD symptoms [91]. The study explored therapist guided music improvisation technique using a variety of musical instruments such as xylophones, maracas, Indian bells, gato drums, djembe, tone bars, guitar, piano and cabassas.

Empirical evidence suggests that music therapy may reduce prominent symptoms of post-traumatic stress, including emotionally dysregulating intrusions, avoidance, negative alterations in mood, and arousal. A study by Zergani & Naderi (2016) demonstrated beneficial effects of Iranian traditional music on quality of life and anxiety symptoms among hospitalized veterans with PTSD [92]. A double blinded randomized control trial, conducted by Pourmovahed et al., (2021) demonstrated that listening to non-verbal music can significantly reduce the severity of PTSD in mothers of premature infants hospitalized in NICU and promote emotional bonding between the mother and baby [93]. The music included the sound of rain, sea, and nature with a slow, gentle, and soothing rhythm. Another randomized controlled trial that examined the effects of music therapy on symptoms of PTSD among prison inmates demonstrated a significant decrease in PTSD-symptoms [94]. A mixed method study examining the efficacy of group drumming therapy in military veterans with PTSD indicated a significant reduction of specific symptoms such as isolation, lack of connectedness, avoidance of traumatic memories, rage, and anxiety [95]. A mixed method study on the feasibility of group music therapy for women with PTSD and complex PTSD found significant changes in the PTSD, dissociation, anxiety, and depression scales, indicating symptom reduction [96]. The qualitative analysis of participant experiences revealed that music assisted in establishing contact with feelings and bodily sensations, as well as providing an experience of expansion, relaxation, and new energy. Furthermore, six participants no longer had a PTSD diagnosis after treatment as shown by the PCL-5 cut-off values, which was sustained even at follow-up.

Music guided relaxation has also shown a positive impact in reducing depression, PTSD and increasing sleep quality in veterans [70]. In a naturalistic study of 102 women with complex PTSD, guided imaginary and classical music significantly reduced symptoms of extreme stress, dissociation, interpersonal problems, and a sense of coherence [97]. Another study demonstrated similar results of guided imaginary and classical music in refugees with PTSD. After sixteen GIM sessions, adult refugees showed positive improvements in PTSD symptoms, sleep quality, well-being, and social functioning [98]. A recent randomized control trial with 74 refugees with PTSD, employing western classical music, New Age music, and music from the participants’ own national culture showed improvements in quality of life and fewer symptoms of psychological dissociation after music therapy [99]. In addition, unlike the standard treatment, the positive effects of music and imagery were manifested even at the 6 month follow-up.

#### 4.2.2. Mechanisms Underlying the Effects of Music in PTSD

Musical stimuli stimulate neural networks associated with various functional domains, such as movement, cognition, communication, emotion, and social responses [100]. Studies clearly demonstrate that instrumental music without lyrics, Chinese and Western music can evoke changes in emotion and stimulate the brain structures involved with motivation, reward, and emotion [101,102,103,104,105]. There is evidence that music can provoke changes in individual emotions, hormone arousal, emotional motor expression, and action movements [102]. The studies included in this analysis used various experimental approaches, such as investigating music-evoked experiences of intense pleasure, emotional responses to consonant or dissonant music, happy or sad music, joy- or fear-evoking music, musical expectancy violations and music-evoked tension. Listening to joyful dance-tunes has been shown to reduce stress and enhance emotional responses, such as joy and peace [105,106]. In particular, music is observed to stimulate increase in blood–oxygen level dependent (BOLD) signals in the amygdala, and the hippocampus [105]. A meta-analysis of functional neuroimaging studies [103] found that the amygdalae, hypothalamus, and hippocampus, which are vital parts of the brain in producing emotion and in experiencing PTSD symptoms, are stimulated by music. None of the included studies used music with lyrics. The ventral tegmental area (VTA), involved in dopamine production and release within the reward system, is also significantly activated by both unfamiliar musical pieces and the participant’s favorite music, in contrast, PFC activity was positively correlated with pleasure scores associated with music [107,108]. However, favorite versus neutral music listening contrasts showed a greater activation in healthy participants than depressed patients [107]. Figure 2 illustrates a summary of brain areas stimulated by music.

PTSD is characterized by hypervigilance associated with altered connectivity between the amygdalae and the hippocampus [41]. Communication between these brain areas is vital in the symptomatology of PTSD. There is evidence that participant’s own favorite music to which they usually had a chill experience, can enhance the connection between the amygdalae, PFC, and the hippocampus [110]. Moreover, contrast analysis of joy, fear and neutral musical stimuli revealed strongest BOLD signals in the superficial amygdala during joyful music, such as classical music, Irish jigs, jazz, reggae, South American and Balkan music [111]. Thus, music could potentially play a role in balancing the processing of stimuli and in reducing the amygdala’s startle response so they can revert to the premorbid state. Moreover, attentive listening to musical clips played with the piano or violin can also stimulate PFC [112], and therefore, can possibly recruit PFC to exert inhibitory control over amygdalic stress responses. Initiating communication between the amygdalae, PFC and hippocampus through music can, therefore, not only mitigate the hypervigilance of PTSD, but can also enhance cognitive processing of emotions.

Attentive listening to or playing music can stimulate neurogenesis and neuroplasticity in the brain [100,106,113] which is relevant for individuals with PTSD who experience neuronal loss and impaired neurogenesis in parts of the limbic system. The increased hippocampal communication with the hypothalamus can also help balance the HPA axis [113]. There is evidence suggesting that musical training in healthy participants can stimulate the hippocampus, induce neurogenesis, and produce a larger hippocampus [114,115,116]. Altering the hippocampal volume can consequently increase positive emotions and regulate negative affect. Koelsch and Skouras (2014) reported increased functional connectivity between the hippocampus and hypothalamus, and amygdalae and nucleus accumbens during exposure to joyful music in healthy adults [117]. The study used non-vocal joyful instrumental music from various epochs and styles. In addition, several studies on music-evoked emotions have reported activity changes in the hippocampus associated with a reduction of emotional stress associated with a lowering of the cortisol level [106,118,119,120]. Overall, 75% of these studies involved experimenter-selected music (classical, new age or easy listening, and world), while the other 25% involved self-selected music, either “entirely self-selected” or “quasi-self-selected”. Clinical studies, which included a majority of the ICU population, demonstrated a stress-reducing effect of music listening irrespective of genre, self-selection of the music, or duration of listening [118]. Classical music demonstrated a significant reduction in cortisol levels among mechanically ventilated ICU survivors [119].

Evidence suggests that active vs. passive music therapy may have differential effects on patient engagement and receptivity. According to fMRI and PET scan studies, active music participation engages more parts of the brain than just listening to music [121]. In a qualitative study, passive music therapy participants reported an immediate therapeutic effect, such as a reduction in anxiety [122]. Active music therapy participants, on the other hand, described interactive session elements as stimulating, alleviating anxiety through pleasant social interaction. Music improvisation (drum based) has been found to be effective in expressing and managing emotions among veterans with PTSD [95]. Moreover, a systematic review showed that passive listening to relaxing music didn’t seem to have any significant effects on PTSD symptoms, suggesting the importance of active music therapy to evoke change in PTSD patients [89]. The researchers posited that specialist skills and an ongoing therapeutic relationship is vital in reducing symptoms of PTSD.

However, music selection needs careful consideration. Music that the participant does not enjoy may result in a stress rather than a relaxation response. Moreover, music can trigger strong memories, which influence the affective response to music and can, therefore, modulate the therapeutic effects of music [123]. In the acute phase of critical illness, despite some controversy around the role of patients’ music preferences, it appears that patient-directed music selection associates with better outcomes [124]. In our review, several studies allowed participants to choose music from a variety of musical genres. However, participants’ choices were restricted within the range of selections offered by the researchers. A systematic review on music interventions for mechanically ventilated patients reported participant dropout rates to be higher in researcher-selected music compared to patient-selected [125]. Instead, studies involving a music therapist to assess patient music preferences have reported no dropouts and a high degree of participant satisfaction [70,98,99]. Therefore, self-selected music appears to be associated with both the effectiveness of music interventions and participant retention.

Basic psychoacoustic properties of music, such as pitch (high or low tone of sounds), rate (fast or slow speed of sounds), loudness (loud or soft intensity of sounds), mode (major or minor key), timbre, and rhythm have been shown to be important factors in the perception and induction of positive as well as negative emotional states. The music therapy research supports music containing a slower tempo, low pitch, containing primarily string composition, regular rhythmic patterns, no extreme changes in dynamics, and no lyrics are associated with relaxation, joy, or peace [126]. The tempo of 60–80 beats per minutes can help induce a state of relaxation and regulate emotions [127]. A study by Beck et al., 2021, used predictable slow-tempo music to decrease arousal and induce relaxation in PTSD patients [99]. In addition, the harmonic complexity of relaxing music should be consonant and remain within the diatonic key with a clear tonal center [126,127]. Predictable music leads to positive responses, such as reward, appraisal, and pleasantness, thus it may support the relaxation response, while dissonant and unexpected harmonies with frequent chord changes activate the amygdala and defeat the purpose of emotion regulation [128]. Research has stated that music with less sharp timbres has been proven to induce relaxation [126]. Possible instrumental arrangements include piano, cello, flute and marimba [127]. In addition, the use of instrumental music over nature sounds can effectively induce relaxation.

As the HPA axis and the production of cortisol play a significant role in PTSD, music can be purposefully used to help regulate the stress response in PTSD. The effects of music on neurobiological aspects of PTSD are summarized in Figure 3.

## 5. Conclusions

PTSD is part of the post-ICU syndrome and impairs the quality of life and functionality of increasing numbers of ICU survivors and their families worldwide. To date, minimal progress has been made in the management of PTSD post-ICU. An impaired HPA axis feedback circle, an exaggerated amygdalic response, hippocampal atrophy and a hypoactive prefrontal cortex accentuate PTSD symptoms, preventing the healthy regulation of the fear response. Music can stimulate the hippocampus and amygdalae that are negatively impacted by stress. Music can also prompt neurogenesis and neuroplasticity within these brain structures and allow the brain to heal. In addition, music can stimulate communication between the amygdalae and PFC/hippocampus, which is vital in the regulation of stress responses. As a result, music can reduce the unnecessary stimulation of the amygdala, increase the effectiveness of both the hippocampus and prefrontal cortex, and balance the HPA-axis. Clinical evidence supports that relaxation music can help improve coping and psychological adaptation after discharge from the ICU. In addition, both sound and music interventions can decrease dissociation symptoms, reduce depression and anxiety levels, and overall reduce the severity of PTSD symptoms. Thus, music could be a cost-effective, easy to access intervention to purposefully address PTSD after an ICU experience and improve the quality of life of both the ICU survivors and their families.

Future studies need to consider those neurobiological effects in planning and testing music interventions for PTSD in survivors of severe illness, such as critical illness. Moreover, elucidating the neurobiological effects of specific types of music, taking into account the intensity of music interventions (i.e., duration of sessions, repetition, length of intervention) and specific patient populations will be important in maximizing the impact of targeted music interventions for post-ICU survivors. Insights into the therapeutic potential of music by determining which types of music (considering individual experiences and preferences) are best suited to stimulate specific limbic brain structures can lead to a more systematic use of music in the therapy of PTSD.

## Figures and Tables

**Figure 1 ijerph-19-03113-f001:**
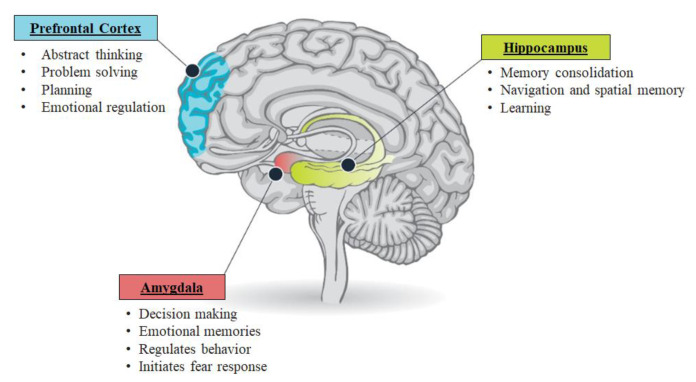
Brain structures involved in emotional regulation and fear response. (Adapted from: Stress and the brain. https://turnaroundusa.org/wp-content/uploads/2020/03/Stress-and-the-Brain_Turnaround-for-Children-032420.pdf accessed on 13 November 2021 [37]).

**Figure 2 ijerph-19-03113-f002:**
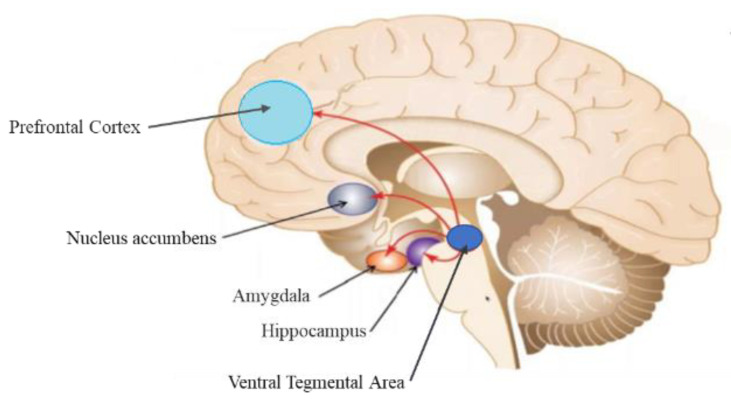
Music stimulates the mesocorticolimbic system. Specifically, it activates the nucleus accumbens, ventral tegmental area, hippocampus, amygdalae, and prefrontal cortex. Adapted from: Desai, R. (2019). Drug addiction. https://drrajivdesaimd.com/wp-content/uploads/2019/06/brain-and-drug-2.jpg accessed on 13 November 2021 [109].

**Figure 3 ijerph-19-03113-f003:**
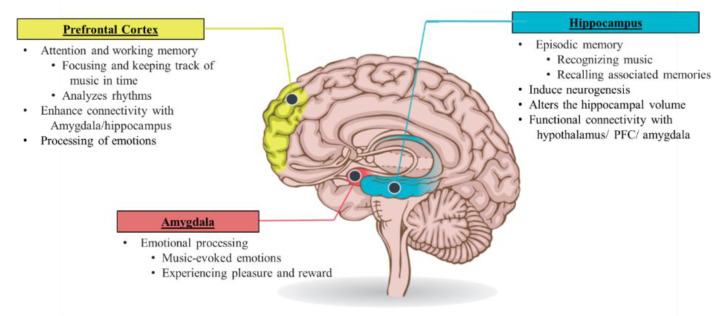
Main effects of music processing in amygdala, hippocampus, and prefrontal cortex. (Adapted from: Cantor, P. (2021). The Stress of This Moment Might Be Hurting Kids’ Development. https://turnaroundusa.org/pamela-cantor-m-d-pens-guest-post-for-education-next/ accessed on 13 November 2021 [129]).

**Table 1 ijerph-19-03113-t001:** Summary of studies employing neuroimaging measurements.

Authors, Date	Imaging Techniques	Brain Measurements
Boccia et. al., 2016	Functional magnetic resonance imaging OR positron emission tomography	Structural brain changes related to PTSD symptomatology Functional connectivity of a brain region
Etkin & Wager, 2007	Functional magnetic resonance imaging OR positron emission tomography	Functional activity of a brain region
Coburn et. al., 2018	Structural magnetic resonance imaging	Structural brain changes
McNerney et. al., 2018	Neuroimaging	Structural brain scan
Postel et. al., 2021	High-resolution magnetic resonance imaging	Structural changes in hippocampal subfields
Gilbertson et. al., 2002	Structural magnetic resonance imaging	Image acquisition and volumetric analyses of hippocampus
Smith et. al., 2005	Magnetic resonance images	Hippocampal volume
van Rooij et. al., 2015	Magnetic resonance imaging	Hippocampal volume
Wang et al., 2010	High-resolution magnetic resonance imaging	Volumes of hippocampal subfields
Grupe et al., 2019	Structural magnetic resonance imaging	Volume of the hippocampus and amygdala
Selemon et. al., 2019	Functional magnetic resonance imaging	Structural and functional changes in brain
Stevens et. al., 2013	Functional magnetic resonance imaging	Functional activity of amygdala and prefrontal cortex Amygdala-prefrontal cortex connectivity
Liu et. al., 2021	3-Tesla magnetic resonance imaging	Functional connectivity of the amygdala and its subregions
Delgado et. al., 2008	Functional magnetic resonance imaging	Functional connectivity and emotional regulation
Johnstone et. al., 2007	Functional magnetic resonance imaging	Functional activity of amygdala and prefrontal cortex
Urry et. al., 2006	Functional magnetic resonance imaging	Brain activity in ventral lateral, dorsolateral, and dorsomedial regions of PFC and amygdala
Xiong et. al., 2013	Event-related functional magnetic resonance imaging	Activity in the inferior frontal cortex, inferior parietal lobule, insula and putamen, posterior cingulate cortex, and amygdala in responses to negative stimuli
Matsuo et. al., 2003	Near-infrared spectroscopy	Hemodynamic response of the prefrontal cortex during a cognitive task
Mary et al., 2020	Functional magnetic resonance imaging	Mechanisms of memory suppression after trauma

**Table 2 ijerph-19-03113-t002:** Summary of PTSD relevant brain areas, structural changes, and effects on behavior.

Brain Areas	Neurobiological Changes	Effects on Behavior
Hippocampus	Reduced volume and activity, reduced dendritic spines and branches of pyramidal neurons in CA3, and Inhibited neurogenesis	Exaggerated activation and inability to terminate stress response, impaired extinction of fear conditioning, non-discrimination between safe/unsafe stimuli, and repressed memories
Amygdala	Increased reactivity, and altered communication with other brain regions	Promotes hypervigilance and impairs discrimination of threat
Prefrontal Cortex (PFC)	Reduced volume and activity, and disrupted communication with amygdalae	Decreased reactivity of PFC to exert inhibitory control over stress responses and dysfunctional thought process and decision making

**Table 3 ijerph-19-03113-t003:** Summary of evidence on the main effects of music interventions in patients experiencing PTSD symptomatology.

Author (s), Year	Study Design	Type of Effect	Measure of Effect	Interpretation of Main Findings
Baker et al., 2018	Systematic Review of 7 interventional studies	Decrease in severity of PTSD	Effect sizes ranged from low-medium effect (PTSD measures used: IES-R and PTSD-8)	Significant reduction in symptoms of PTSD when there was ongoing therapist involvement compared to when there was little therapist or no therapist involvement.
Story & Beck, 2017	Mixed methods	Improved coping Improved emotional regulation Decrease in severity of PTSD	Change in PTSD symptoms, ES = 1.0	Participants reported experiencing music as a tool for coping with PTSD symptoms, regulating emotions, decreasing arousal, expressing repressed feelings, and connecting with others.
Pourmovahed et al., 2021	Randomized control trial	Improved emotional regulation Decrease in severity of PTSD Decreased anxiety levels	Severity of the PTSD decreased significantly after the intervention in the experimental group (F 1, 57 = 1046, *p* = 0.003) Difference between the two groups (F1, 07 = 1058, *p* < 0.03) confirmed significant effect of the non-verbal music on decreasing the PTSD severity	Listening to non-verbal music reduced severity of PTSD and the mother’s stress consequently promoting emotional bonding between the mother and baby.
Bensimon et al., 2008	Mixed method	Improved emotional regulation Decreased anxiety levels	Reducing the client’s self-reported anxiety during confrontation with feared stimuli Effect measures not reported	Coping with difficulties such as feelings of loneliness, harsh traumatic memories, outbursts of anger, and loss of control.
Carr et al., 2012	Mixed method study	Decrease in severity of PTSD Decrease in depression	IES-R significant reduction from baseline of (−17.20; 95% CI: [−24.94, −9.45; *p* = 0.0012]) Reduction in BDI-II symptom severity (−0.71)	Music and guided imagery can improve symptoms of Complex PTSD and dissociation, alleviate interpersonal problems, and enhance factors that promote health.
Rudstam et al., 2017	Mixed method study	Decrease in severity of PTSD Decrease in depression Decreased anxiety levels Decreased dissociation symptoms Improved quality of life	Pre-post Comparisons PCL-5, ES = 1.10 DES-T, ES = 0.85 DES, ES = 1.00 HSC25-I, ES = 1.17 HSC25-II, ES = 0.58 PSOM, ES = 0.24 Follow-up PCL-5, ES = 1.49 DES, ES = 0.92 DES-T, ES = 1.10 HSC25-I, ES = 1.35 HSC25-II, ES = 0.74 PSOM, ES = 0.59	Significant decreases in PTSD symptoms with very large effect sizes, and dissociation with large effect sizes, and an increase in quality of life with small to medium effect size. Music helped establish contact with feelings and body sensations and provided an experience of expansion, relaxation, and new energy.
Maack, 2012	Mixed method study	Decrease in severity of PTSD Decreased dissociation symptoms Improved quality of life	Kruskal–Wallis-Test shows that there was a significant difference in change of severity of symptoms between the groups (*p* < 0.001). KW test statistic not reported. Mann–Whitney Tests shows that there was a significant difference in change of severity of symptoms between the GIM and the control group (U = 1.50, *p* < 0.001).	The symptoms of the participants of the GIM group improved significantly more than the symptoms of the participants of the PITT group.
Beck et al., 2017	Pre- post-test study	Decrease in severity of PTSD Improved quality of life	Pre-post Comparisons HTQ subscales, ES = 1.17 Avoidance, ES = 1.11 Intrusion, ES = 1.03 Hypervigilance, ES = 0.60 WHO-5 Wellbeing scale, ES = 0.18	Significant changes in positive directions on all four outcome measures, PTSD symptoms, sleep quality, well-being, and social functioning.
Macfarlane et al., 2019	Pre- post-test study	Decrease in severity of PTSD	Average reduction of PTSD symptoms of 38% between the entrance screening and the final point of the intervention, using PSS-I	A drop of ten points or more on PSS-I score for eight of the participants, among which five had a final scored below PTSD threshold. Applicable in a complex clinical setting with a very mixed and treatment resistant population, who were not eligible for EMDR or another type of trauma treatment, at the moment of enrollment.
Blanaru et al., 2012	Mixed method study	Decrease in depression	Significant reduction in BDI score for depression following music relaxation compared with baseline [F (1,11) = 14.8, *p*< 0.003]	Music relaxation was found to be effective and led to significant improvements in sleep measures and significant reduction of depression score in PTSD patients.
Beck et al., 2021	Randomized control trial	Decreased dissociation symptoms Improved quality of life	Music group well-being, large effect size ES = 0.58, *p* = 0.005 at end of treatment ES = 0.61, *p* = 0.004 at follow-up Psychoform dissociation, small to large effect size ES = 0.35 at end of treatment ES = 0.71, *p* = 0.0002 at follow-up Psychological treatment groupwell-being, small effect size ES = 0.06 at end of treatment small ES = 0.18 at follow-up Psychoform dissociation, medium effect size ES = 0.31 at end of treatment ES = 0.41 at follow-up	Small to large effect sizes in both psychological treatment group and music therapy group, with significant medium effect sizes, for well-being and psychoform dissociation at follow-up. A high dropout rate of 40% occurred in the psychological treatment group, compared to 5% in the music therapy group.
Zergani & Naderi, 2016	Randomized control trial	Decreased anxiety levels Improved quality of life	Significant difference between experiment and control groups for anxiety symptoms (F-13.67; *p* < 0.0001), STAI scale, and quality of life (F-26.99; *p* < 0.0001), SF-36 scale	The effect of music remained stable even after one month of follow-up.

PCL-5: PTSD Checklist for DSM-5; DES: Dissociative Experience Scale; DES-T: Dissociative Experience Scale Taxon; HSCL-25: Hopkins Symptom Checklist; PSOMS: Positive State of Mind Scale; IES-R: Impact of Event Scale–Revised; PTSD-8: Post-traumatic Stress Disorder 8-item; BDI: Beck Depression Inventory, HTQ: Harvard Trauma Questionnaire; STAI: State-Trait Anxiety Inventory; SF-36: Short Form Health Survey is a 36-item; WHO-5: WHO Well-being Index; ES: Effect Size using Cohen’s d.

## Data Availability

Not applicable.

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
