# Peer review of "A Neurobiological Framework for the Therapeutic Potential of Music and Sound Interventions for Post-Traumatic Stress Symptoms in Critical Illness Survivors"

_ijerph, 2022, doi:10.3390/ijerph19053113_

Round 1

Reviewer 1 Report

The review of the literature proposed by Pant et al. and entitled "A neurobiological framework for the therapeutic potential of music and sound interventions for post-traumatic stress symptoms in critical illness survivors” offers an interesting and fairly complete synthesis of the literature on this issue, with some figures concerning the neurophysiological mechanisms involved (illustrations adapted from previous articles). If this review has the merit of clarity and being quite concise, two types of additional information could certainly make the manuscript more interesting and distinguish it from other papers already published on the same issue. First of all, it would be useful that the authors summarize (with an additional Table) the methodologies most often used in the literature in order to notably know the number of studies that have made neuroimaging measurements and to specify the imaging techniques (EEG, MRI, others...) and the brain measurements (structural, functional...). Then, the authors should further discuss the differential benefits of listening or practicing music. For the moment the effects of listening or practice are mixed and reported without distinguishing them as being able to impact the different types of deleterious mechanisms present in patients with PSTD, as presented at the beginning of the manuscript. However, can we expect the same benefits from listening to or practicing music? It would certainly be necessary to go further than what the authors currently propose, because there are certainly some answers in the literature.

Some small points of improvement: 1-There are elements underlined in table 2 without our understanding why. 2-Although the cited literature is widespread, there are important articles that are not cited, for example (Mary et al. Resilience after trauma: The role of memory suppression, 2020, Science) which would notably complete the references 40 concerning the sentence p. 5, line 223 "Thus, the hypoactivity of the prefrontal cortex can explain some of the symptoms of PTSD such as the inability to focus, solve problems and guide thoughts or emotions using working memory". 

Author Response

Dear reviewers,

Thank you very much for the thorough review of our paper. Your review has helped us improve and clarify our manuscript. We greatly appreciate the comprehensive and insightful review. We responded to your comments with appropriate amendments, which are underlined in text for ease of reference. Please see a detailed list of responses to specific comments, along with the pages where amendments can be found, below. Again, thank you for the opportunity to revise our manuscript. 

Kind regards

Response to comments from Reviewer # 1:

Comment

Response

The review of the literature proposed by Pant et al. and entitled "A neurobiological framework for the therapeutic potential of music and sound interventions for post-traumatic stress symptoms in critical illness survivors” offers an interesting and fairly complete synthesis of the literature on this issue, with some figures concerning the neurophysiological mechanisms involved (illustrations adapted from previous articles). If this review has the merit of clarity and being quite concise, two types of additional information could certainly make the manuscript more interesting and distinguish it from other papers already published on the same issue.

Thank you very much for the feedback.

First of all, it would be useful that the authors summarize (with an additional Table) the methodologies most often used in the literature in order to notably know the number of studies that have made neuroimaging measurements and to specify the imaging techniques (EEG, MRI, others...) and the brain measurements (structural, functional...).

Thank you for the suggestion. A new table has been added pages 3 and 4 of the manuscript and attached at the end for your reference (Table 1). 

The authors should further discuss the differential benefits of listening or practicing music. For the moment the effects of listening or practice are mixed and reported without distinguishing them as being able to impact the different types of deleterious mechanisms present in patients with PSTD, as presented at the beginning of the manuscript. However, can we expect the same benefits from listening to or practicing music? It would certainly be necessary to go further than what the authors currently propose, because there are certainly some answers in the literature.

Thank you for the suggestion. This has been added on page 15 line 407-418 (underlined text). 

There are elements underlined in table 2 without our understanding why.

Thank you for pointing out that this has been fixed.

Although the cited literature is widespread, there are important articles that are not cited, for example (Mary et al. Resilience after trauma: The role of memory suppression, 2020, Science) which would notably complete the references 40 concerning the sentence p. 5, line 223 "Thus, the hypoactivity of the prefrontal cortex can explain some of the symptoms of PTSD such as the inability to focus, solve problems and guide thoughts or emotions using working memory". 

Thank you for directing us to this interesting study.

Response to comments from Reviewer # 2:

Comment

Response

The article discusses an interesting and important topic of addressing PTSD following critical illness and ICU treatment.  Specifically, it explores the use of music-based interventions which may provide an effective and cost-effect treatment for PTSD.  This article can contribute to the sparse literature on this topic.

The article was well written, with the background need, methodology, and neurobiological framework well defined and clearly explained.  Table 1 was a helpful addition to summarize brain areas and engagement in PTSD.

Thank you very much for the feedback.

Section 4.2.2

Regarding the description and literature review supporting the use of music, two important aspects should be included:

1.            The importance of client-preferred music (rather than researcher chosen) on the emotional response of the client.

You do mention this briefly, but this should be highlighted a more.  The personal significance of autobiographical associations to a piece of music can further strengthen the emotional response to, and impact of, the music.

2.            The impact of specific musical components on emotion.

 For example, melody’s contour, predictability, and range can support relaxation, tempo can support reduction in anxiety due to lowering of breathing and/or heart rate, and harmony (dissonance vs Consonance) can impact emotions by expressing tension or not.

I recommend that you add a paragraph describing how the components of music impact emotions, improve mood, and reduce stress or anxiety.  Your literature review and examples in section 4.2.2 will provide further examples and evidence of the effectiveness of music.

Thank you very much for these comments. Section 4.2.2 has been updated on page 15 lines 419-450 (underlined text). We now mention the importance of client-preferred music, the role of music in triggering memories and related affective responses, and the impact of specific musical components

Well done and interesting.  This article can inform the development and implementation of the use of music for post-ICU PTSD.  The above 2 points will further strengthen your theoretical rationale

Thank you very much for the suggestion.

Response to comments from Reviewer # 3:

Comment

Response

The paper reviews the intersection of the neurobiological evidence on PTSD and the literature on the effects of music therapy in neuropsychiatric conditions. It gives a contribution to a significant field of future research on post ICU PTSD treatment.

Thank you very much for the feedback.

The paper abides by SANRA guideline for the assessment of critical narrative review reporting. For this reason and since the Authors claim that the aim of the paper is to outline an 'evidence based' theoretical framework, one would expect that measures of the effects listed on Table 2 had been reported, if provided by some papers of the referenced literature. That would improve the presentation of the review data and the soundness of the scientific point the Authors make.

Thanks for the comment. The table has been updated on page 9-14 of the manuscript and attached at the end for your reference.

Language and style are fine, but a check is needed for a possible typo of underscores in Table 2 and line 358.

Thank you for the comment. The typo has been fixed.

Table 1. Summary of studies employing neuroimaging measurements

Authors, date

Imaging techniques

Brain measurements

Boccia, et. al., 2016

Functional magnetic resonance imaging OR positron emission tomography

Structural brain changes related to PTSD symptomatology

Functional connectivity of a brain region

Etkin & Wager, 2007

Functional magnetic resonance imaging OR positron emission tomography

Functional activity of a brain region

Coburn, et. al., 2018

Structural magnetic resonance imaging

Structural brain changes

McNerney, et. al., 2018

Neuroimaging

Structural brain scan

Postel, et. al., 2021

High-resolution magnetic resonance imaging

Structural changes in hippocampal subfields

Gilbertson, et. al., 2002

Structural magnetic resonance imaging

Image acquisition and volumetric analyses of hippocampus

Smith, et. al., 2005

Magnetic resonance images

Hippocampal volume

van Rooij, et. al., 2015

Magnetic resonance imaging

Hippocampal volume

Wang, et al., 2010

High-resolution magnetic resonance imaging

Volumes of hippocampal subfields

Grupe, et al., 2019

Structural magnetic resonance imaging

Volume of the hippocampus and amygdala

Selemon, et. al., 2019

 Functional magnetic resonance imaging

Structural and functional changes in brain

Stevens, et. al., 2013

Functional magnetic resonance imaging

Functional activity of amygdala and prefrontal cortex

Amygdala- prefrontal cortex connectivity

Liu, et. al., 2021

3-Tesla magnetic resonance imaging

Functional connectivity of the amygdala and its subregions.

Delgado, et. al., 2008

Functional magnetic resonance imaging

Functional connectivity and emotional regulation

Johnstone, et. al., 2007

Functional magnetic resonance imaging

Functional activity of amygdala and prefrontal cortex

Urry, et. al., 2006

Functional magnetic resonance imaging

Brain activity in ventral lateral, dorsolateral, and dorsomedial regions of PFC and amygdala

Xiong, et. al., 2013

Event-related functional magnetic resonance imaging

Activity in the inferior frontal cortex, inferior parietal lobule, insula and putamen, posterior cingulate cortex, and amygdala in responses to negative stimuli

Matsuo, et. al., 2003

Near-infrared spectroscopy

Hemodynamic response of the prefrontal cortex during a cognitive task

Mary et al., 2020

Functional magnetic resonance imaging

Mechanisms of memory suppression after trauma

Table 3. Summary of evidence on the main effects of music interventions in patients experiencing PTSD symptomatology.

Author (s), year

Study design

Type of effect

Measure of effect

Interpretation of Main findings

Baker et al., 2018

Systematic Review of 7 interventional studies

Decrease in severity of PTSD

Effect sizes ranged from low-medium effect (PTSD measures used: IES-R and PTSD-8)

Significant reduction in symptoms of PTSD when there was ongoing therapist involvement compared to when there was little therapist or no therapist involvement.

Story & Beck, 2017

Mixed methods

Improved coping

Improved emotional regulation

Decrease in severity of PTSD

Change in PTSD symptoms, ES=1.0

Participants reported experiencing music as a tool for coping with PTSD symptoms, regulate emotions, decrease arousal, express repressed feelings, and connect with others.

Pourmovahed et al., 2021

Randomized control trial

Improved emotional regulation

Decrease in severity of PTSD

Decreased anxiety levels

Severity of the PTSD decreased significantly after the intervention in

the experimental group (F 1,57 = 1046, p = 0.003)

Difference between the two groups (F1,07 = 1058, p < 0.03), confirmed significant effect of the non‑verbal music on decreasing the PTSD severity

Listening to non‑verbal music reduced severity of PTSD and the mother’s stress consequently promoting emotional bonding between the mother and baby.

Bensimon et al., 2008

Mixed method

Improved emotional regulation

Decreased anxiety levels

Reducing the client’s self-reported anxiety during confrontation with feared stimuli

Effect measures not reported

Coping with difficulties such as feelings of loneliness, harsh traumatic memories, outbursts of anger, and loss of control.

Carr et al., 2012

Mixed method study

Decrease in severity of PTSD

Decrease in depression

IES-R significant reduction from baseline of (−17.20; 95%CI: [−24.94, −9.45; p = .0012])

Reduction in BDI-II symptom severity (−0.71)

Music and guided imagery can improve symptoms of Complex PTSD and dissociation, alleviate interpersonal problems, and enhance factors that promote health.

Rudstam et al., 2017

Mixed method study

Decrease in severity of PTSD

Decrease in depression

Decreased anxiety levels

Decreased dissociation symptoms

Improved quality of life

 Pre-post

Comparisons

·        PCL-5, ES=1.10

·         DES-T, ES =0.85

·        DES, ES=1.00

·        HSC25-I, ES=1.17

·        HSC25-II, ES=0.58

·        PSOM, ES=0.24

Follow-up

·        PCL-5, ES=1.49

·        DES, ES=0.92

·        DES-T, ES =1.10

·        HSC25-I, ES=1.35

·        HSC25-II, ES=0.74

·        PSOM, ES=0.59

Significant decreases in PTSD symptoms with very large effect sizes, and dissociation with large effect sizes, and an increase in quality of life with small to medium effect size. Music helped establish contact with feelings and body sensations and provided an experience of expansion, relaxation, and new energy.

Maack, 2012

Mixed method study

Decrease in severity of PTSD

Decreased dissociation symptoms

Improved quality of life

Kruskal-Wallis-Test shows that there was a significant difference in change of severity of symptoms between the groups (p < .001). KW test statistic not reported.

Mann-Whitney Tests shows that there was a significant difference in change of severity of symptoms between the GIM and the control group (U = 1.50, p < .001).

The symptoms of the participants of the GIM group improved significantly more than the symptoms of the participants of the PITT group.

Beck et al., 2017

Pre-posttest study

Decrease in severity of PTSD

Improved quality of life

Pre-post

Comparisons

·        HTQ subscales, ES=1.17

·        Avoidance, ES = 1.11

·        Intrusion, ES =1.03

·        Hypervigilance, ES =0.60

·        WHO-5 Wellbeing scale, ES =0.18

Significant changes in positive directions on all four outcome measures, PTSD symptoms, sleep quality, well-being, and social functioning.

Macfarlane et al., 2019

Pre-posttest study

Decrease in severity of PTSD

Average reduction of PTSD symptoms of 38% between the entrance screening and the final point of the intervention, using PSS-I

A drop of ten points or more on PSS-I score for eight of the participants. Among which five had a final scored below PTSD threshold.

Applicable in a complex clinical setting with a very mixed and treatment resistant population, who were not eligible for EMDR or another type of trauma treatment, at the moment of enrollment.

Blanaru et al., 2012

Mixed method study

Decrease in depression

Significant reduction in BDI score for depression following music relaxation compared with baseline

[F (1,11) =14.8, P<0.003]

Music relaxation was found to be effective and led to significant improvements in sleep measures and significant reduction of

depression score in PTSD patients.

Beck et al., 2021

Randomized control trial

Decreased dissociation symptoms

Improved quality of life

Music group

well-being, large effect size

·        ES = 0.58, p = .005 at end of treatment

·        ES =0.61, p = .004 at Follow up

Psychoform dissociation, small to large effect size

·        ES= 0.35 at end of treatment

·        ES=0.71, p = .0002 at Follow up

Psychological treatment group,

 well-being, small effect size

·         ES= 0.06 at end of treatment

·        small ES = 0.18 at follow up

Psychoform dissociation, medium effect size

·        ES= 0.31 at end of treatment

·         ES=0.41 at Follow up

Small to large effect sizes in both psychological treatment group and music therapy group, with significant medium effect sizes, for well-being and psychoform dissociation at follow-up.

A high dropout rate of 40% occurred in the psychological treatment group,

compared to 5% in the music therapy group.

Zergani & Naderi, 2016

Randomized control trial

Decreased anxiety levels

Improved quality of life

Significant difference between experiment and control groups for anxiety symptoms (F - 13.67; p < 0.0001), STAI scale, and quality of life (F - 26.99; p < 0.0001), SF-36 scale

The effect of music remained stable even after one month of follow-up.

PCL-5: PTSD Checklist for DSM-5; DES: Dissociative Experience Scale; DES-T: Dissociative Experience Scale Taxon; HSCL-25: Hopkins Symptom Checklist; PSOMS: Positive State of Mind Scale; IES-R:  Impact of Event Scale–Revised; PTSD-8: Post-traumatic Stress Disorder 8-item; BDI: Beck Depression Inventory, HTQ: Harvard Trauma Questionnaire; STAI: State-Trait Anxiety Inventory; SF-36: Short Form Health Survey is a 36-item; WHO-5: WHO Well-being Index; ES: Effect Size using Cohen’s d

Reviewer 2 Report

A neurobiological framework for the therapeutic potential of music and sound interventions for post-traumatic stress symptoms in critical illness survivors.

REVIEW:

The article discusses an interesting and important topic of addressing PTSD following critical illness and ICU treatment.  Specifically, it explores the use of music-based interventions which may provide an effective and cost-effect treatment for PTSD.  This article can contribute to the sparse literature on this topic.

The article was well written, with the background need, methodology, and neurobiological framework well defined and clearly explained.  Table 1 was a helpful addition to summarize brain areas and engagement in PTSD. 

Section 4.2.2

Regarding the description and literature review supporting the use of music, two important aspects should be included:

  1. The importance of client-preferred music (rather than researcher chosen) on the emotional response of the client.

You do mention this briefly, but this should be highlighted a more.  The personal significance of autobiographical associations to a piece of music can further strengthen the emotional response to, and impact of, the music. 

I recommend a few more sentences mentioning this, plus reference.

  1. The impact of specific musical components on emotion.  

For example, melody’s contour, predictability, and range can support relaxation, tempo can support reduction in anxiety due to lowering of breathing and/or heart rate, and harmony (dissonance vs constance ) can impact emotions by expressing tension or not.

You mention how a range of musical styles can influence emotions (265-266) and specific instruments (373-373).  However, it is the musical components within these styles (plus the associated emotions to the style) and that are being performed (rather than the instrument that the song is performed on) that are impacting emotions. However, certain instruments (their timbre) can be more relaxing than others.  Thus, piano or harp may be more relaxing than percussive instruments.

I recommend that you add a paragraph describing how the components of music impact emotions, improve mood, and reduce stress or anxiety.  Your literature review and examples in section 4.2.2 will  provide further examples and evidence of the effectiveness of music.

Well done and interesting.  This article can inform the development and implementation of the use of music for post-ICU PTSD.  The above 2 points will further strengthen your theoretical  rationale.

Author Response

Dear reviewers,

Thank you very much for the thorough review of our paper. Your review has helped us improve and clarify our manuscript. We greatly appreciate the comprehensive and insightful review. We responded to your comments with appropriate amendments, which are underlined in text for ease of reference. Please see a detailed list of responses to specific comments, along with the pages where amendments can be found, below. Again, thank you for the opportunity to revise our manuscript. 

Kind regards

Response to comments from Reviewer # 1:

Comment

Response

The review of the literature proposed by Pant et al. and entitled "A neurobiological framework for the therapeutic potential of music and sound interventions for post-traumatic stress symptoms in critical illness survivors” offers an interesting and fairly complete synthesis of the literature on this issue, with some figures concerning the neurophysiological mechanisms involved (illustrations adapted from previous articles). If this review has the merit of clarity and being quite concise, two types of additional information could certainly make the manuscript more interesting and distinguish it from other papers already published on the same issue.

Thank you very much for the feedback.

First of all, it would be useful that the authors summarize (with an additional Table) the methodologies most often used in the literature in order to notably know the number of studies that have made neuroimaging measurements and to specify the imaging techniques (EEG, MRI, others...) and the brain measurements (structural, functional...).

Thank you for the suggestion. A new table has been added page 3 and 4 of the manuscript and attached at the end for your reference (Table 1). 

The authors should further discuss the differential benefits of listening or practicing music. For the moment the effects of listening or practice are mixed and reported without distinguishing them as being able to impact the different types of deleterious mechanisms present in patients with PSTD, as presented at the beginning of the manuscript. However, can we expect the same benefits from listening to or practicing music? It would certainly be necessary to go further than what the authors currently propose, because there are certainly some answers in the literature.

Thank you for the suggestion. This has been added page 15 line 407-418 (underlined text). 

There are elements underlined in table 2 without our understanding why.

Thank you pointing that this has been fixed.

Although the cited literature is widespread, there are important articles that are not cited, for example (Mary et al. Resilience after trauma: The role of memory suppression, 2020, Science) which would notably complete the references 40 concerning the sentence p. 5, line 223 "Thus, the hypoactivity of the prefrontal cortex can explain some of the symptoms of PTSD such as the inability to focus, solve problems and guide thoughts or emotions using working memory". 

Thank you for directing to this interesting study.

Response to comments from Reviewer # 2:

Comment

Response

The article discusses an interesting and important topic of addressing PTSD following critical illness and ICU treatment.  Specifically, it explores the use of music-based interventions which may provide an effective and cost-effect treatment for PTSD.  This article can contribute to the sparse literature on this topic.

The article was well written, with the background need, methodology, and neurobiological framework well defined and clearly explained.  Table 1 was a helpful addition to summarize brain areas and engagement in PTSD.

Thank you very much for the feedback.

Section 4.2.2

Regarding the description and literature review supporting the use of music, two important aspects should be included:

1.            The importance of client-preferred music (rather than researcher chosen) on the emotional response of the client.

You do mention this briefly, but this should be highlighted a more.  The personal significance of autobiographical associations to a piece of music can further strengthen the emotional response to, and impact of, the music.

2.            The impact of specific musical components on emotion.

 For example, melody’s contour, predictability, and range can support relaxation, tempo can support reduction in anxiety due to lowering of breathing and/or heart rate, and harmony (dissonance vs Consonance) can impact emotions by expressing tension or not.

I recommend that you add a paragraph describing how the components of music impact emotions, improve mood, and reduce stress or anxiety.  Your literature review and examples in section 4.2.2 will provide further examples and evidence of the effectiveness of music.

Thank you very much for these comments. Section 4.2.2 has been updated page 15 line 419-450 (underlined text). We now mention the importance of client-preferred music, the role of music in triggering memories and related affective responses, and the impact of specific musical components

Well done and interesting.  This article can inform the development and implementation of the use of music for post-ICU PTSD.  The above 2 points will further strengthen your theoretical rationale

Thank you very much for the suggestion.

Response to comments from Reviewer # 3:

Comment

Response

The paper reviews the intersection of the neurobiological evidence on PTSD and the literature on the effects of music therapy in neuropsychiatric conditions. It gives a contribution to a significant field of future research on post ICU PTSD treatment.

Thank you very much for the feedback.

The paper abides by SANRA guideline for the assessment of critical narrative review reporting. For this reason and since the Authors claim that the aim of the paper is to outline an 'evidence based' theoretical framework, one would expect that measures of the effects listed on Table 2 had been reported, if provided by some papers of the referenced literature. That would improve the presentation of the review data and the soundness of the scientific point the Authors make.

Thanks for the comment. Table have been updated page 9-14 of manuscript and attached at the end for your reference.

Language and style are fine, but a check is needed for a possible typo of underscores in Table 2 and line 358.

Thank you for the comment. The typo has been fixed.

Table 1. Summary of studies employing neuroimaging measurements

Authors, date

Imaging techniques

Brain measurements

Boccia, et. al., 2016

Functional magnetic resonance imaging OR positron emission tomography

Structural brain changes related to PTSD symptomatology

Functional connectivity of a brain region

Etkin & Wager, 2007

Functional magnetic resonance imaging OR positron emission tomography

Functional activity of a brain region

Coburn, et. al., 2018

Structural magnetic resonance imaging

Structural brain changes

McNerney, et. al., 2018

Neuroimaging

Structural brain scan

Postel, et. al., 2021

High-resolution magnetic resonance imaging

Structural changes in hippocampal subfields

Gilbertson, et. al., 2002

Structural magnetic resonance imaging

Image acquisition and volumetric analyses of hippocampus

Smith, et. al., 2005

Magnetic resonance images

Hippocampal volume

van Rooij, et. al., 2015

Magnetic resonance imaging

Hippocampal volume

Wang, et al., 2010

High-resolution magnetic resonance imaging

Volumes of hippocampal subfields

Grupe, et al., 2019

Structural magnetic resonance imaging

Volume of the hippocampus and amygdala

Selemon, et. al., 2019

 Functional magnetic resonance imaging

Structural and functional changes in brain

Stevens, et. al., 2013

Functional magnetic resonance imaging

Functional activity of amygdala and prefrontal cortex

Amygdala- prefrontal cortex connectivity

Liu, et. al., 2021

3-Tesla magnetic resonance imaging

Functional connectivity of the amygdala and its subregions.

Delgado, et. al., 2008

Functional magnetic resonance imaging

Functional connectivity and emotional regulation

Johnstone, et. al., 2007

Functional magnetic resonance imaging

Functional activity of amygdala and prefrontal cortex

Urry, et. al., 2006

Functional magnetic resonance imaging

Brain activity in ventral lateral, dorsolateral, and dorsomedial regions of PFC and amygdala

Xiong, et. al., 2013

Event-related functional magnetic resonance imaging

Activity in the inferior frontal cortex, inferior parietal lobule, insula and putamen, posterior cingulate cortex, and amygdala in responses to negative stimuli

Matsuo, et. al., 2003

Near-infrared spectroscopy

Hemodynamic response of the prefrontal cortex during a cognitive task

Mary et al., 2020

Functional magnetic resonance imaging

Mechanisms of memory suppression after trauma

Table 3. Summary of evidence on the main effects of music interventions in patients experiencing PTSD symptomatology.

Author (s), year

Study design

Type of effect

Measure of effect

Interpretation of Main findings

Baker et al., 2018

Systematic Review of 7 interventional studies

Decrease in severity of PTSD

Effect sizes ranged from low-medium effect (PTSD measures used: IES-R and PTSD-8)

Significant reduction in symptoms of PTSD when there was ongoing therapist involvement compared to when there was little therapist or no therapist involvement.

Story & Beck, 2017

Mixed methods

Improved coping

Improved emotional regulation

Decrease in severity of PTSD

Change in PTSD symptoms, ES=1.0

Participants reported experiencing music as a tool for coping with PTSD symptoms, regulate emotions, decrease arousal, express repressed feelings, and connect with others.

Pourmovahed et al., 2021

Randomized control trial

Improved emotional regulation

Decrease in severity of PTSD

Decreased anxiety levels

Severity of the PTSD decreased significantly after the intervention in

the experimental group (F 1,57 = 1046, p = 0.003)

Difference between the two groups (F1,07 = 1058, p < 0.03), confirmed significant effect of the non‑verbal music on decreasing the PTSD severity

Listening to non‑verbal music reduced severity of PTSD and the mother’s stress consequently promoting emotional bonding between the mother and baby.

Bensimon et al., 2008

Mixed method

Improved emotional regulation

Decreased anxiety levels

Reducing the client’s self-reported anxiety during confrontation with feared stimuli

Effect measures not reported

Coping with difficulties such as feelings of loneliness, harsh traumatic memories, outbursts of anger, and loss of control.

Carr et al., 2012

Mixed method study

Decrease in severity of PTSD

Decrease in depression

IES-R significant reduction from baseline of (−17.20; 95%CI: [−24.94, −9.45; p = .0012])

Reduction in BDI-II symptom severity (−0.71)

Music and guided imagery can improve symptoms of Complex PTSD and dissociation, alleviate interpersonal problems, and enhance factors that promote health.

Rudstam et al., 2017

Mixed method study

Decrease in severity of PTSD

Decrease in depression

Decreased anxiety levels

Decreased dissociation symptoms

Improved quality of life

 Pre-post

Comparisons

·        PCL-5, ES=1.10

·         DES-T, ES =0.85

·        DES, ES=1.00

·        HSC25-I, ES=1.17

·        HSC25-II, ES=0.58

·        PSOM, ES=0.24

Follow-up

·        PCL-5, ES=1.49

·        DES, ES=0.92

·        DES-T, ES =1.10

·        HSC25-I, ES=1.35

·        HSC25-II, ES=0.74

·        PSOM, ES=0.59

Significant decreases in PTSD symptoms with very large effect sizes, and dissociation with large effect sizes, and an increase in quality of life with small to medium effect size. Music helped establish contact with feelings and body sensations and provided an experience of expansion, relaxation, and new energy.

Maack, 2012

Mixed method study

Decrease in severity of PTSD

Decreased dissociation symptoms

Improved quality of life

Kruskal-Wallis-Test shows that there was a significant difference in change of severity of symptoms between the groups (p < .001). KW test statistic not reported.

Mann-Whitney Tests shows that there was a significant difference in change of severity of symptoms between the GIM and the control group (U = 1.50, p < .001).

The symptoms of the participants of the GIM group improved significantly more than the symptoms of the participants of the PITT group.

Beck et al., 2017

Pre-posttest study

Decrease in severity of PTSD

Improved quality of life

Pre-post

Comparisons

·        HTQ subscales, ES=1.17

·        Avoidance, ES = 1.11

·        Intrusion, ES =1.03

·        Hypervigilance, ES =0.60

·        WHO-5 Wellbeing scale, ES =0.18

Significant changes in positive directions on all four outcome measures, PTSD symptoms, sleep quality, well-being, and social functioning.

Macfarlane et al., 2019

Pre-posttest study

Decrease in severity of PTSD

Average reduction of PTSD symptoms of 38% between the entrance screening and the final point of the intervention, using PSS-I

A drop of ten points or more on PSS-I score for eight of the participants. Among which five had a final scored below PTSD threshold.

Applicable in a complex clinical setting with a very mixed and treatment resistant population, who were not eligible for EMDR or another type of trauma treatment, at the moment of enrollment.

Blanaru et al., 2012

Mixed method study

Decrease in depression

Significant reduction in BDI score for depression following music relaxation compared with baseline

[F (1,11) =14.8, P<0.003]

Music relaxation was found to be effective and led to significant improvements in sleep measures and significant reduction of

depression score in PTSD patients.

Beck et al., 2021

Randomized control trial

Decreased dissociation symptoms

Improved quality of life

Music group

well-being, large effect size

·        ES = 0.58, p = .005 at end of treatment

·        ES =0.61, p = .004 at Follow up

Psychoform dissociation, small to large effect size

·        ES= 0.35 at end of treatment

·        ES=0.71, p = .0002 at Follow up

Psychological treatment group,

 well-being, small effect size

·         ES= 0.06 at end of treatment

·        small ES = 0.18 at follow up

Psychoform dissociation, medium effect size

·        ES= 0.31 at end of treatment

·         ES=0.41 at Follow up

Small to large effect sizes in both psychological treatment group and music therapy group, with significant medium effect sizes, for well-being and psychoform dissociation at follow-up.

A high dropout rate of 40% occurred in the psychological treatment group,

compared to 5% in the music therapy group.

Zergani & Naderi, 2016

Randomized control trial

Decreased anxiety levels

Improved quality of life

Significant difference between experiment and control groups for anxiety symptoms (F - 13.67; p < 0.0001), STAI scale, and quality of life (F - 26.99; p < 0.0001), SF-36 scale

The effect of music remained stable even after one month of follow-up.

PCL-5: PTSD Checklist for DSM-5; DES: Dissociative Experience Scale; DES-T: Dissociative Experience Scale Taxon; HSCL-25: Hopkins Symptom Checklist; PSOMS: Positive State of Mind Scale; IES-R:  Impact of Event Scale–Revised; PTSD-8: Post-traumatic Stress Disorder 8-item; BDI: Beck Depression Inventory, HTQ: Harvard Trauma Questionnaire; STAI: State-Trait Anxiety Inventory; SF-36: Short Form Health Survey is a 36-item; WHO-5: WHO Well-being Index; ES: Effect Size using Cohen’s d

Reviewer 3 Report

The paper reviews the intersection of the neurobiological evidence on PTSD and the literature on the effects of music therapy in neuropsychiatric conditions. It gives a contribution to a significant field of future research on post ICU PTSD treatment.

The paper abides by SANRA guideline for the assessment of critical narrative review reporting. For this reason and since the Authors claim that the aim of the paper is to outline an 'evidence based' theoretical framework, one would expect that measures of the effects listed on Table 2 had been reported, if provided by some papers of the referenced literature. That would improve the presentation of the review data and the soundness of the scientific point the Authors make.

Language and style are fine, but a check is needed for a possible typo of underscores in Table 2 and line 358.

Author Response

(The authors gave the same response as above.)
